# Beneficial Effects of Anti-Inflammatory Diet in Modulating Gut Microbiota and Controlling Obesity

**DOI:** 10.3390/nu14193985

**Published:** 2022-09-26

**Authors:** Soghra Bagheri, Samaneh Zolghadri, Agata Stanek

**Affiliations:** 1Medical Biology Research Center, Health Technology Institute, Kermanshah University of Medical Sciences, Kermanshah 6714415185, Iran; 2Department of Biology, Jahrom Branch, Islamic Azad University, Jahrom 7414785318, Iran; 3Department of Internal Medicine, Angiology and Physical Medicine, Faculty of Medical Sciences in Zabrze, Medical University of Silesia, Batorego 15 St, 41-902 Bytom, Poland

**Keywords:** dietary intervention, inflammation, insulin resistance, obesity, anti-inflammatory diets, gut microbiota

## Abstract

Obesity has consistently been associated with an increased risk of metabolic abnormalities such as diabetes, hyperlipidemia, and cardiovascular diseases, as well as the development of several types of cancer. In recent decades, unfortunately, the rate of overweight/obesity has increased significantly among adults and children. A growing body of evidence shows that there is a relationship between metabolic disorders such as obesity and the composition of the gut microbiota. Additionally, inflammation is considered to be a driving force in the obesity–gut microbiota connection. Therefore, it seems that anti-inflammatory nutrients, foods, and/or diets can play an essential role in the management of obesity by affecting the intestinal flora and controlling inflammatory responses. In this review, we describe the links between the gut microbiota, obesity, and inflammation, and summarize the benefits of anti-inflammatory diets in preventing obesity.

## 1. Introduction

Obesity is a complex multifactorial disease that is directly related to the chance of developing insulin resistance and an increased risk of diabetes, dyslipidemia, hypertension, cardiovascular diseases [1,2], and other metabolic disorders [3]. With growing concerns about obesity and related diseases, there has been scientific interest in understanding the mechanisms associated with obesity and exploring innovative management strategies to control it. Mounting evidence suggests a strong link between obesity and high-fat diets as well as alterations in intestinal microbiota [4,5,6]. The normal human gut microbiota is an ecological community of trillions of commensal [7], and symbiotic micro-organisms [8] that play an essential role in maintaining metabolic function and immune system homeostasis [9]. Although the exact mechanism by which the gut microbiota contributes to obesity is largely unknown, it appears that obesity-induced changes in the intestinal microbiota are accompanied by significant alterations in gut permeability and short-chain fatty acid (SCFA) content [10]. Moreover, evidence shows that disruption of the gut microbiota activates the adaptive and innate immunity of the intestine and increases the inflammatory level by displacing immunogenic bacterial products. Although inflammation is a normal and automatic defense response of the body against foreign pathogen invasion, excessive inflammatory response results in a series of diseases, such as obesity. “Obese microbiota” indirectly affects hypothalamic gene expression and promotes overeating by increasing systemic inflammation and microglial activation, affecting vagus nerve activity [10,11]. Inflammation of adipose tissue will encourage angiogenesis. Angiogenesis is a critical event in tissue remodeling and in the development of obesity. By elevating the inflammatory cytokines in circulation, energy expenditure is systemically promoted. The inflammatory response also induces insulin resistance and hyperglycemia [12] (Figure 1). Accordingly, the timely regulation of inflammation is vital to controlling obesity and its complications.

The gut microbiota has been proposed to be one of the critical factors regulating early events, causing inflammation associated with obesity and metabolic dysfunction. Thus, considering the effect of gut microbiota composition on inflammatory responses in the host [13] and inducing obesity, targeting the gut microbiota by means of anti-inflammatory diets (also known as the hypoallergenic diet, elimination diet, and oligoantigenic diet [14]) could be beneficial in combatting inflammation-related diseases such as obesity and obesity-related diseases. Bolte et al. found that certain foods with anti-inflammatory effects play an essential role in regulating gut microbiota and maintaining health [15]. In this review, we attempt to discuss several aspects related to the association of gut microbiota and inflammation with obesity on the basis of the most recently published studies. In addition, we discuss the relationship between the main inflammatory and pro-inflammatory stimuli with obesity, and finally we summarize the latest scientific evidence supporting anti-inflammatory dietary intervention in the regulation of the gut microbiota and obesity control.

## 2. Methods

An extensive search was conducted online using the PubMed/MEDLINE, Scopus, and Google Scholar databases to retrieve the most relevant studies including randomized controlled trials (RCTs), in vivo and in vitro experimental studies, and literature reviews and meta-analyses in recent years. The main keywords were: obesity, inflammation, adipokines, cytokines, SCFA, gut microbiota, polyphenols, probiotics, and anti-inflammatory diet.

## 3. Relationships between Gut Microbiota and Inflammation

Inflammation is an automatic defense response of the body to foreign pathogen invasion. However, excessive inflammatory response leads to a host of diseases such as obesity, cardiovascular disorders, neurodegenerative manifestations, and cancer. Research has confirmed the direct nexus between gut microbiota alteration (dysbiosis) and inflammatory conditions such as inflammatory bowel disease and obesity [16], immune-mediated inflammatory diseases (e.g., ulcerative colitis, Crohn’s disease, rheumatoid arthritis, and multiple sclerosis) [17], and type I and II diabetes mellitus [18]. The gut microbiota–inflammation interplay depends on leading host-related factors such as diet mode, age, sex, body mass index (BMI), host genotype, and antibiotic use. Dysbiosis is a critical contributor to chronic inflammatory and metabolic dysfunction, as reviewed elsewhere [19]. It is worth noting that SCFAs, mainly acetate, propionate, and butyrate, produced by the microbial fermentation of undigested food, play a crucial role in colonic health by various mechanisms. According to van der Beek 2017, SCFAs maintain metabolic homeostasis in colonocytes through anti-inflammatory and anticarcinogenic effects [20]. In a non-systematic review by Alsharairi (2021), the therapeutic role of SCFAs derived from infant intestinal microbiota in obesity was discussed in detail [21].

Several studies have shown that macrophages [22,23], epithelial cells [24], neutrophils [25], and monocytes [26] are modulated by SCFAs [27].

## 4. Obesity and Inflammation

Research has found a potential link between obesity and inflammatory markers, which suggests that a persistent inflammatory response is a potential risk factor [28]. In obese individuals, excessive accumulation of subcutaneous and/or abdominal fat induces adipose tissue inflammation. The changes that occur in the adipose tissue as a result of obesity are complex. Increased lipid storage results in adipose tissue dysfunction including hypoxia, adipocyte hypertrophy, and increased cell death. Adipose tissue is the main fat storage depot and the largest endocrine organ, regulating several physiological and pathological processes by systemically secreting adipokines and pro- and anti-inflammatory cytokines [29], including leptin (a 16-kDa adipokine), adiponectin (a 28 KDa protein), resistin (a 12 kDa adipocytokine), visfatin (a 52 kDa protein product of the pancreatic beta cell growth factor gene), omentin and chemerin, etc. [30]. The M2-like adipose tissue macrophages (ATMs), distributed within the adipose, perform diverse physiological functions, including inhibiting proliferation of adipocyte progenitors, promoting dead adipocyte clearance, and secreting anti-inflammatory cytokines such as interleukin (IL)-1Rα (IL-1Rα), IL-4, IL-10, and IL-13. Adipose tissue dysfunction promotes a microenvironment in which the adipocytes begin to secrete pro-inflammatory cytokines, including IL-6, IL-8, monocyte chemoattractant protein (MCP-1) and tumor necrosis factor-α (TNF-α). MCP-1 and other chemokines produced by adipocytes and immune cells promote increased infiltration of circulating monocytes and other innate and adaptive immune cells into the adipose tissue [31]. The secretion of anti-inflammatory and protective cytokines is stopped in obese people, while the secretion of inflammatory cytokines and adipokines, which act systemically or locally to induce peripheral insulin resistance, is started [32,33] (Figure 2).

Leptin is a pro-inflammatory product of the obesity gene [34], which is produced predominantly by differentiated white adipose tissue and acts in the brain to regulate energy homeostasis. In the acute phase of inflammation, leptin stimulates the release of cytokines such as interleukin-1β (IL-1β) and promotes T cell polarization toward pro-inflammatory T helper 1 (TH1) response [35]. Leptin promotes activation of monocytes, macrophages, and natural killer (NK) cells [36], and regulates various functions in the immune and endocrine systems, including hematopoiesis, osteogenesis, angiogenesis, wound healing, and inflammation [37]. Moreover, it plays a central role in glucose homeostasis, energy balance, and regulation of caloric expenditure and food intake. Leptin signaling is mediated through Janus kinase 2, SH2-containing protein tyrosine phosphatase 2 (SHP2)/mitogen-activated protein kinase (MAPK), insulin receptor substrate (IRS)/phosphatidylinositol 3 kinase (PI3K), signal transducer and activator of transcription 3 (STAT3), and 5′ adenosine monophosphate-activated protein kinase (AMPK)/acetyl-CoA carboxylase (ACC) [38,39]. There is a direct correlation between fat mass and leptin levels in obese patients. Leptin stimulates inflammatory responses by binding to the long isoform of the leptin receptor, expressed by different immune cell types. At the same time, certain infectious and inflammatory stimuli, such as lipopolysaccharide (LPS), IL1, and TNF-α, can also increase leptin levels, which correlate with the chronic inflammation in obesity. Therefore, there is a bidirectional interaction between leptin and inflammation [40]. Adiponectin is an anti-inflammatory cytokine that suppresses monocyte adhesion to endothelial cells [41] and inhibits macrophage transformation into foam cells [42]. In human dendritic cells and macrophages, this cytokine directly induces the synthesis of anti-inflammatory (IL-10 and IL-1RA) cytokines [43] and decreases pro-inflammatory cytokines (TNFα, IL-1β, IL-6, IL-8) [44], thereby suppressing inflammation. Studies have reported that adiponectin reduces insulin resistance in muscle and fat and improves insulin sensitivity.

In obese individuals, it has been reported that production of adiponectin is drastically reduced [45]. Moreover, resistin, which is produced by peripheral inflammatory cells of blood, macrophages, and monocytes, could establish a metabolic link between obesity and insulin resistance. Visfatin is synthesized predominantly by adipocytes and macrophages, hepatocytes, and neutrophils. This adipokine stimulates the synthesis and storage of triacylglycerols in adipocytes. It exerts its biological function through the insulin receptor. In obese patients, increased visfatin levels, similarly to increased adiponectin levels, may play a protective role. Omentin and chemerin are other adipokines that potentiate insulin-dependent glucose uptake by adipocytes. They are also associated with obesity-induced insulin resistance. Chemerin is also believed to be a link between obesity and inflammation. Its levels are particularly high in very obese subjects suffering from severe obesity [30].

Obesity-related factors such as diets, host genetics, gut microbiota, and other related factors may all contribute to the initiation and maintenance of adipose tissue inflammation [5]. It has been recognized that adipocytes secrete a variety of inflammatory molecules. In addition to adipocytes, inflammatory cells, including macrophages, neutrophils, B cells, T cells, and others, release the majority of inflammatory molecules in the adipose tissue of obese humans [46]. A large body of evidence points to systemic inflammation in obese individuals being mainly characterized by increased accumulation and inflammatory polarization of innate and adaptive immune cells in adipose tissue, skeletal muscle, liver, intestine, pancreatic islets, and brain, and may prompt the development and progression of obesity-related metabolic disorders leading to insulin resistance, β-cell dysfunction, and type 2 diabetes through multiple metabolic signaling pathways [47,48]. To date, several studies have reported the role of inflammatory cells in adipose tissue and analyzed its potential role in insulin resistance associated with obesity. Wu et al. published a comprehensive review on inflammation in adipose tissue and its potential role in insulin resistance associated with obesity. They also discussed the role of primary inflammatory cells in adipose tissue. A deep understanding of the mechanisms of inflammatory responses in overweight/obese individuals could provide unique opportunities to adopt appropriate intervention strategies to reduce the risk of obesity-related comorbidities [49].

## 5. Obesity and Gut Microbiota

Recently, an increasing number of studies have suggested that an imbalance in microbial populations and microbial diversity (dysbiosis) is significantly associated with several disorders, including obesity, diabetes, and cardiometabolic diseases, as well as behavioral and neurological complications [50,51,52,53]. The alteration of the homeostatic balance of the gut microbiota results in changes in downstream metabolites in the obese. To date, several mechanisms have been proposed to explain the potential role of the gut microbiota in the development of obesity. Some evidence suggests that the gut microbiota contributes to metabolism and metabolic disorders through a close cross-talk with host adipose tissue [54,55].

Among them, the fermentation of non-digestible dietary polysaccharides by intestinal bacteria results in the production of SCFAs, which can induce lipogenesis in the liver and triglyceride accumulation in host adipocytes [56]. SCFAs can bind to the G-protein-coupled receptors (GPRs) and activate downstream signaling pathways. Notably, SCFAs inhibit histone deacetylases (HDACs) in mucosal immune cells and colonocytes [20]. HDACs are epigenetic enzymes that activate or suppress the target gene, depending on the modification position [57]. Based on these studies, butyrate is the most potent HDAC inhibitor in colon cancer cells [58]. Butyrate activates the AhR pathway by increasing AhR recruitment to the target gene promoter through HDAC inhibition. The AhR pathway is involved in some metabolic and immune processes that are vital for gut homeostasis and optimal coexistence of the host and its microbiome [59]. SCFA might also indirectly induce insulin sensitivity through increased systemic levels of gut-derived glucagon-like peptide 1 (GLP-1) [60].

Moreover, in vitro studies have demonstrated that SCFAs stimulate leptin expression in adipocytes through activation of free fatty acid receptor 3 (FFAR3) [61]. Aberrations in leptin expression are one of the most frequent features in the onset and progression of obesity [62]. *Coprobacillus cateniformis* and *Clostridium leptum* are positively correlated with an adipocyte-derived hormone, namely, leptin [63]. Leptin centrally suppresses food intake and increases energy expenditure through its receptor in the hypothalamus [64]. Additionally, certain lactic acid bacteria can convert glutamate to gamma-aminobutyric acid (GABA), which then expresses 5-hydroxytryptamine and GABA-binding proteins to control appetite [65]. These findings highlight a new line of communication between the gut microbiota and the brain. Mounting evidence indicates that the gut microbiota might target the brain via a direct pathway of electrical stimulation of the vagus nerve and an indirect means involving an immune–neuroendocrine mechanism [66]. Torres-Fuentes et al. [67] reviewed and discussed the role of the gut–brain axis in obesity. They mentioned that food harvesting, appetite, and energy balance are regulated by an intricate network of neuroendocrine signals that provide a reciprocal link between the gastrointestinal tract and the brain. They also emphasized that the obesity-associated microbiota leads to fat deposition, insulin resistance, and inflammation by altering the host’s metabolic machinery, balance of energy uptake, food reward signals, and central appetite.

## 6. The Interplay of the Gut Microbiota, Inflammation, and Obesity

The most recent studies comparing the gut microbiome in adults with obesity and control groups have demonstrated that gut microbiota diversity in obese people is significantly decreased, and there are significant differences between the microbiota of these two groups at different levels, from phylum to species. Furthermore, it has been shown that obese people have abnormalities mainly with respect to carbohydrate and lipid metabolism pathways compared to the control group [68,69,70,71]. Additionally, comparing the structure of the gut microbiota between obese and normal school-age children revealed that the gut microbiota in obese children was less diverse than in the control group. In addition, a significant difference in the relative frequency of gut microbiota was observed at different classification levels [72]. On the other hand, there is evidence indicating that the composition of the gut microbiome can enhance dietary energy intake and subsequently contribute to the obese phenotype [73]. A dietary fiber that cannot be entirely hydrolyzed by human enzymes during digestion is catabolized by gut microbiota, with SCFAs being the main product of this process [74]. SCFAs can affect the proper function of tight junctions between epithelial cells and subsequently regulate the absorption of xenobiotics. It has also been proved that these molecules can affect appetite, immune system, and blood pressure regulation, as well as lipid and glucose metabolism [73,75].

Some studies have reported that the dominant bacteria living in the intestine belong to four bacterial phylotypes: *Bacteroidetes*, *Firmicutes*, *Proteobacteria*, and *Actinobacteria*. *Bacteroidetes* and the *Firmicutes* make up more than 90% of the gut microbiota [76]. Although some reports have suggested that the gut microbiota in healthy people has a high ratio of *Bacteroidetes* to *Firmicutes*, and this ratio is reversed in obese individuals, with a lower frequency of *Bacteroidetes* [70,77,78,79], other reports have not found this relationship [69,80]. Some studies have even observed an inverse relationship [68,81].

A meta-analytical study in 2014 also found no association between the ratio of *Bacteroidetes/Firmicutes* and obesity [82]. Nevertheless, later reports have still reported the relationship between *Firmicutes/Bacteroidetes* ratio and obesity. For instance, a 2018 study of pregnant women indicated that *Firmicutes* levels and the *Firmicutes/Bacteroidetes* ratio were higher in overweight and obese individuals [83]. Finally, a more recent meta-analysis in 2019 reported that the *Firmicutes/Bacteroidetes* ratio was not a reproducible marker in human cohorts on the basis of an examination of body mass index (BMI) [84]. In addition, investigations into the relationship between the gut microbiome and childhood obesity have reached conflicting findings. Indeed, these investigations have mainly used BMI to measure obesity. However, a new study in this field demonstrates that the gut microbiome and SCFAs are significantly related to body fat distribution in the pediatric population [85]. Additionally, another survey of female adolescents showed that the abundance of *Firmicutes* is associated with waist and neck circumference, but not with BMI or body fat percentage. Moreover, this study indicated that girls with a higher waist circumference had higher concentrations of inflammatory markers, leptin, and high-sensitivity C-reactive protein (hs-CRP) [86].

An investigation conducted on colorectal cancer patients reported that obesity did not have a significant effect on the diversity and abundance of intestinal bacteria of colorectal cancer patients. Still, the profile of the intestinal microbiota of obese patients revealed a decrease in butyrate-producing bacteria and an overabundance of opportunistic pathogens, which in turn could be at least partially responsible for harmful bacterial metabolites, higher levels of the pro-inflammatory cytokine IL-1β, and intestinal permeability in these patients [87]. The results of a study on pregnant women with gestational diabetes showed that the gut microbiome is different in women who are resistant and sensitive to nutritional therapy, so characteristic gut microbiomes that are able to improve blood glucose and microbiomes that are able to increase blood glucose were significantly increased in susceptible and resistant groups, respectively [88].

## 7. Influence of Anti-Inflammatory Diet on Gut Microbiota, Insulin Signaling, and Obesity

Typically, anti-inflammatory diets include the consumption of unrefined and minimally processed foods, nutrients including fiber, mono- and poly unsaturated fatty acids, lean protein sources such as chicken, and various spices, and reducing the consumption of red meat, high-fat dairy foods, and saturated and trans fats that have notably been shown to diminish inflammation [89,90]. The effect of an anti-inflammatory diet on weight loss has been reported many times. For instance, a randomized controlled trial (RCT) study examining the metabolic level and inflammatory status of 81 participants indicated that an anti-inflammatory diet led to a significant decrease in body weight and visceral adipose tissue and improved the cardiometabolic and inflammatory status of the participants [91]. Moreover, a cross-sectional study on 535 adolescent boys found that adherence to a diet with a low inflammatory index score was related to a reduction in body fat percentage [92].

Various studies have demonstrated that gut microbiome diversity is influenced by diet [93,94], and that the microbiome is directly involved in regulating pro-inflammatory and anti-inflammatory responses in the gut [15]. A systematic 2022 review of RCTs in adults, including 18 trials on 1385 obese individuals who were subjected to several dietary interventions, including standard and healthy diets, revealed that diets modified the microbiota signature in a macronutrient composition-dependent manner, so that the percentage of carbohydrates, fats, and fibers seemed to be the primary regulator of the composition, richness, and function of the microbiome, while the increase in protein had relatively few effects on the species and genera. Standard diets, including low-carbohydrate, low-fat, or Mediterranean diets, lead to lower weight and BMI, while other healthy diets promoting whole grains, fruits, and vegetables may or may not be associated with weight loss [95]. In the following sections, the effect of anti-inflammatory diets on the gut microbiome and obesity will be discussed. Due to the difficulty of conducting controlled dietary intervention trials in humans, most relevant studies have been conducted on animals. However, in the following sections, the authors have attempted to summarize and discuss the human studies.

### 7.1. Mediterranean Diet

The Mediterranean diet includes consumption of plant foods (high amounts), olive oil (the primary source of fat), dairy products (mainly cheese and yogurt), low to moderate amounts of fish and chicken, zero to four eggs per week, low amounts of red meat, and minor to moderate amounts of wine, typically consumed with meals [96]. Recently, a systematic review of RCTs reported that the Mediterranean diet could produce significant differences in interleukins-1α, -1β, -4, -5, -6, -7, -8, -10, and -18, IFN-γ, TNF-α, CRP, and hs-CRP compared to the control diet [97]. Findings on the association between following a Mediterranean diet and the risk of overweight/obesity have been conflicting; therefore, a meta-analytical study was conducted in 2022 showing that following a Mediterranean diet is inversely associated with the risk of overweight and/or obesity [98].

There is a lot of evidence supporting the beneficial effects of the Mediterranean diet on gut microbiota in preventing obesity. Several reports have elucidated that the consumption of whole grains, which have been reported to reduce inflammation and body weight, is accompanied by a change in the intestinal microbiota in terms of species and gender [99,100,101,102,103]. Examination of the microbiome of 1425 individuals in four cohort studies, including individuals with Crohn’s disease, ulcerative colitis, irritable bowel syndrome, and healthy subjects, revealed that consumption of processed foods and animal-derived foods were related to higher frequencies of *Firmicutes* and *Ruminococcus* species and endotoxin synthesis pathways. On the other hand, it was clarified that plant foods and fish were positively associated with SCFA-producing bacteria and nutrient metabolism pathways [15]. Less adherence to the Mediterranean diet was associated with higher *Firmicutes–Bacteroidetes* ratio, while better adherence to the Mediterranean diet was associated with a greater abundance of *Christensenellaceae* and higher SCFA levels [104]. A systematic review study in 2020 on the impact of dairy products and derivatives on the intestinal microbiota demonstrated that the consumption of dairy products (milk, yogurt, and kefir) enhanced the abundance of beneficial genera Lactobacillus and *Bifidobacterium*, while consumption of dairy derivatives (whey and casein) and the amount of dairy consumed did not change the composition of the gut microbiota [105]. An investigation on 360 Spanish adults revealed that high adherence to the Mediterranean diet increased the abundance of certain health-related bacterial species, with the species *Bifidobacterium animalis* showing the strongest association. In addition, this study found that some SCFA-producing bacteria were also associated with the Mediterranean diet [106]. However, a 2022 systematic review of RCTs and observational studies on the effects of the Mediterranean diet on gut microbiota and microbial metabolites indicated that, overall, there is no clear evidence of a consistent effect of the Mediterranean diet on gut microbiota composition or metabolism [107].

### 7.2. Dietary Fiber

In general, dietary fiber comprises soluble and insoluble carbohydrates from plant sources that are not digested by human enzymes, are not absorbed in the small intestine, and have positive effects on human health [108]. There has been extensive research on the relationship between dietary fiber and obesity, the results of which show a significant effect of fiber consumption in reducing obesity [108,109,110]. On the other hand, it has been demonstrated that the intake of dietary fiber changes the composition and activity of gut microbiota [111,112]. Both epidemiological and randomized controlled studies have confirmed that receiving a greater abundance of nutritional fiber is associated with a reduction in the risk of Type 2 diabetes [113]. Some evidence has revealed that the impact of dietary fiber consumption on improving insulin sensitivity is due to an increase in the colonic production of the SCFAs acetate, propionate, and butyrate, the final products of the fermentation of dietary fiber by the gut microbiota [114,115]. Indeed, the anti-inflammatory effects of high circulating concentrations of SCFAs achieved by consuming high-fiber diets have been demonstrated in mouse models. A similar human study discovered that both acetate and propionate were significantly increased in people who consumed a high-fiber diet compared to the low-fiber group, but no changes in peripheral blood inflammatory factors were observed in the five-day intervention; this might be due to the short intervention time [116]. A comparison of gut microbiota and SCFAs in American women and Ghanaian women (who consumed notably more dietary fiber) indicated that lean Ghanaians had a greater abundance of microbial genes that catalyze the production of butyric acid through the fermentation of pyruvate or branched-chain amino acids. Meanwhile, in obese Ghanaians and American women (regardless of BMI), a higher frequency of microbial genes related to enzymes related to the fermentation of amino acids such as alanine, aspartate, lysine, and glutamate was observed [117].

### 7.3. Polyphenols

Polyphenols are secondary plant metabolites that are probably the most abundant antioxidants in daily food intake. Due to their high molecular weight and complex structure, a small percentage of dietary polyphenols is absorbed within the digestive tract. Still, in the large intestine, they are converted into bioactive phenolic metabolites with low molecular weight by the intestinal microbiota [118]. Various clinical trials have been performed on the impact of polyphenols on intestinal microbiota and obesity, revealing the positive effect of these components in terms of changing the gut microbiota to reduce obesity [119]. More recent findings also confirm the previous results. For instance, in a double-blind RCT on 62 girls with obesity aged 6–10 years old for 12 weeks, decaffeinated green tea polyphenols were shown to have a significant effect on alleviating obesity and delaying early sexual development [120]. A randomized, double-blind, parallel clinical trial on 72 obese and overweight volunteers over 16 weeks suggested that polyphenol supplementation significantly reduced body fat mass [121]. The results of a controlled clinical trial demonstrated that daily consumption of oranges has a positive effect on the composition and activity of intestinal microbiota, such that it has increased the population of *Bifidobacterium* and *Lactobacillus* species, and increased SCFA production. In addition, it improved metabolic biomarkers such as low-density lipoprotein (LDL)-cholesterol, glucose, and insulin sensitivity [122]. An RCT study reported that consumption of genistein (from the flavonoid family) for two months in obese people caused a decrease in insulin resistance, a reduction in metabolic endotoxemia, an increase in the phosphorylation of 5′-adenosine monophosphate-activated protein kinase, and an increase in the oxidation of skeletal muscle fatty acids (via expressing genes involved in fatty acid oxidation) by causing changes in the intestinal microbiota, especially by means of an increase in the *Verrucomicrobia* phylum [123].

Polyphenols are an essential component of fiber-containing foods and should be considered when studying the role of fiber-containing foods in human health [124]. A diet rich in fiber that was enriched with polyphenols and vegetable protein, significantly changed the fecal microbiota in patients with type 2 diabetes compared to the control group. It increased the number of *Faecalibacterium prausnitzii* and *Akkermansia muciniphila*, two types of bacteria known to have anti-inflammatory effects, while it decreased the number of *Prevotella copri*. Along with these changes, a significant decrease in the biomarkers of glucose, triglycerides, total and LDL-cholesterol, HbA1c, CRP, and FFAs, and an increase in antioxidant activity were reported for the intervention group [125]. In a dietary intervention on mildly hypercholesterolaemic subjects, it was shown that olive pomace (rich in polyphenols and fibers) helped. However, while it did not affect the diversity of the fecal microbiota, it caused a decreasing trend of lactobacilli and *Ruminococcus* and an increasing trend of *Bifidobacteria* in the microbiota of the participants. In addition, a considerable increase in the metabolic output of the intestinal microbiota of these subjects was observed, which is probably due to these changes [126]. The results of an RCT in subjects with high cardiometabolic risk confirmed that diets rich in polyphenols increased the diversity of the dominant fecal bacteria in these individuals, and these changes were connected to changes in glucose/lipid metabolism [127].

### 7.4. Omega-3-Rich Diet

Long-chain polyunsaturated fatty acids (LC) are fatty acids with 18 or more carbons, and are classified into two leading families, ω3 (LCn3) and ω6 (LCn6), depending on the position of the first double bond from the methyl end group of the fatty acid [128]. A 2017 meta-analysis of RCTs found that omega-3 supplementation did not effectively reduce body weight, but reduced waist circumference and triglyceride levels in overweight and obese adults. However, due to the small number and poor quality of the RCTs available in the meta-analysis, it was suggested that more studies be conducted in this field [129]. Another meta-analysis study of RCTs investigating the effects of supplementation with omega-3 fatty acids on metabolic status in pregnant women reported an increase in serum high-density lipoprotein (HDL)-cholesterol concentration and a decrease in CRP in the participants [130]. A case study of a healthy 45-year-old man found that an omega-3-rich diet reduced species diversity in the subject’s microbiota but increased the diversity of several butyrate-producing bacteria. In fact, this effect on the microbiota was observed at the genus/species level instead of at the phylum level [131]. Another study on 22 healthy participants found that high doses of omega-3 supplements did not cause significant changes in the gut microbiome at the phylum level in the volunteers. However, they caused a reversible increase in the abundance of several SCFA-producing genera [132]. Some findings from an investigation of 47 healthy adults showed that ω-3 fatty acid supplementation had little effect on the gut microbiota. However, increased colonic bacterial diversity after ω-3 fatty acid supplementation was significantly predictive of decreased colonic prostaglandin E2 concentrations [133].

The results of an RCT in subjects with high cardiometabolic risk reported that diets rich in LCn3 affected the composition of the gut microbiota in these people. It has been shown that the diversity of dominant fecal bacteria decreased after low-LCn3 and polyphenols and High-LCn3 diets, and these changes were connected to changes in glucose/lipid metabolism [127]. Another study on 126 participants with borderline hypercholesterolemia indicated that during the dietary intervention, the relative abundance of a single species, *Clostridium leptum*, increased dramatically, and along with that, various plasma markers of the metabolic health status improved, including apolipoprotein B, triglyceride, and the ratio of total cholesterol to HDL. In this study, butyrate production was discussed as a possible process for improving blood lipid profile [134].

### 7.5. Probiotics

Probiotics are non-pathogenic microorganisms that reach the gut in an active state and have potential benefits for the host [135]. Various reports have described the beneficial effects of probiotics on factors related to inflammation and obesity, as well as their impact on the intestinal microbiota [136,137]. *Bifidobacterium* and *lactobacillus* (probiotic yogurt) caused significant decreases in the serum levels of IL-1β, TNF-α, and CRP, as well as considerable increases in the serum levels of IL-6 and IL-10 in the intervention group [138]. A double-blind RCT in obese women demonstrated that a low-energy diet per se induces changes in metabolite profiles related to reduced inflammation and positive effects on body weight. In addition, it showed that both probiotics and synbiotics produce changes in metabolites associated with improved metabolic health. Finally, it suggested that synbiotics (due to specific metabolite changes) along with a low-energy diet could be more beneficial than probiotics or diet alone [139]. A synbiotic is a mixture of living microorganisms and substrate(s) selectively used by host microorganisms that is beneficial to host health [140]. The results of a 2021 meta-analysis of RCTs indicated that probiotic consumption significantly reduced total cholesterol, triglycerides, and LDL-cholesterol levels [141]. An RCT reported in 2022 revealed that supplementation of the studied probiotics significantly improved the function of the intestinal barrier, leading to a significant reduction in concentrations of lipopolysaccharides (LPS). It also substantially increased HDL-cholesterol and SCFA levels (propionic and butyric acid) and improved obesity-related biomarkers [142]. Treatment with multispecies probiotics changed the biochemical, physiological, and immunological parameters of the gut microbiota in postmenopausal women with obesity but did not affect diversity or taxonomic classification [143]. In contrast, the use of probiotics and the Mediterranean diet in overweight breast cancer survivors resulted in a significant increase in the number and diversity of bacterial species in the group that consumed probiotics. In addition, the ratio of *Bacteroidetes/Firmicutes* decreased in the intervention group, while it increased in the control group. Apart from this, the parameters body weight, BMI, fasting glucose, and insulin resistance exhibited significant decreases in both groups, but in the intervention group, waist circumference, waist/hip ratio, and fasting insulin has also revealed a considerable decline [144].

## 8. Conclusions

In conclusion, it is well known that obesity is associated with chronic inflammation and insulin resistance, and the gut microbiota is an important part of overcoming obesity. Moreover, it appears that obesity is inclined toward a pro-inflammatory condition by increasing the content of inflammatory mediators, and the intestinal microbiota contributes to the progression of this condition. Therefore, increasing attention has been focused on targeting the gut microbiota for the treatment of metabolic diseases such as obesity. Among various treatments, anti-inflammatory diets are thought to have the ability to reverse or regulate intestinal flora disorders and to improve health status. Accordingly, anti-inflammatory dietary intervention in obese individuals may produce promising results through improved insulin resistance and metabolic functions. However, knowledge about anti-inflammatory approaches is still limited, and future studies are required in order to explore the potential of targeting inflammation in specific organs/tissues to treat obesity-related metabolic disorders. We hope that this review can provide a new perspective for the development of new treatments for obesity-linked metabolic disease.

## Figures and Tables

**Figure 1 nutrients-14-03985-f001:**
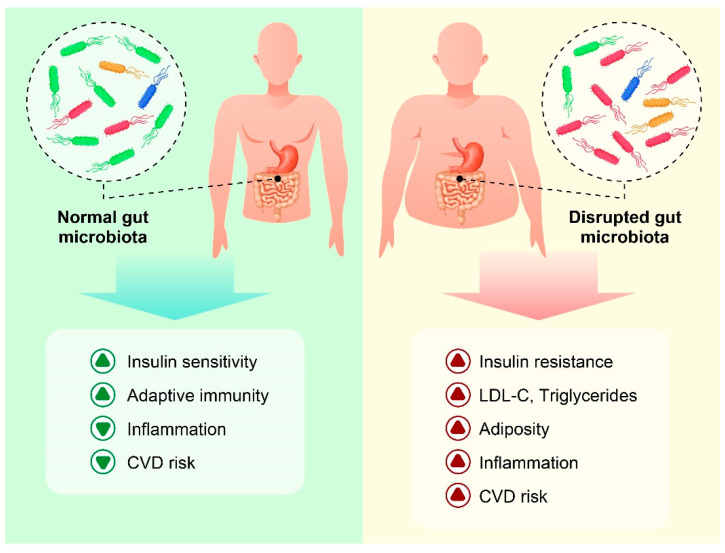
Normal gut microbiota and maintenance of proper metabolic function in healthy individuals (**left**). The role of dysbiosis in the progression of some metabolic disorders in obese individuals (**right**).

**Figure 2 nutrients-14-03985-f002:**
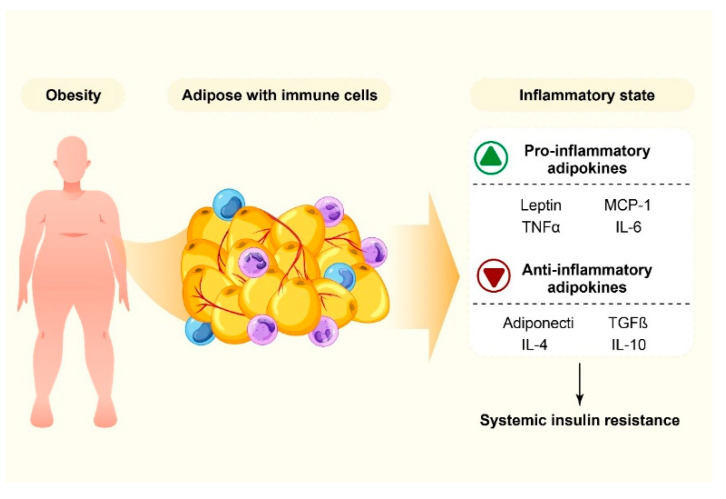
The interplay of inflammation and obesity.

## Data Availability

No new data were created or analyzed in this study. Data sharing does not apply to this article.

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
