# Peer review of "Beneficial Effects of Anti-Inflammatory Diet in Modulating Gut Microbiota and Controlling Obesity"

_nutrients, 2022, doi:10.3390/nu14193985_

Round 1
Reviewer 1 Report
In this review, authors discussed the interaction of obesity, inflammation and obesity, summarized the studies in human investigating diet/diet supplements effect on inflammation, microbiota and obesity. It is of high interest to readers since effective dietary products may be idea option for long term obesity management. There are several concerns need to be addressed.
In section “4. Obesity and Inflammation”, authors discussed the “gut-brain axis” rather than “obesity and inflammation”. Please check carefully and provide the correct information.
In addition, in this section (and others), authors summarized gut brain interaction from one review paper. Please consider providing critical original research publications as references.
Reviewer 2 Report
· Please include information about the search strategy conducted in abstract (e.g., we searched on PubMed/Medline, Web of Science ….etc).
· Line 19: Suggest- it seems that anti-inflammatory nutrients, foods, and/or diets……
· Please add “obesity”, “anti-inflammatory diets” and “gut microbiota” to the keywords list.
· Line 32-37: Authors should state that the gut microbiota contains both pathogenic and commensal bacteria, and the alteration of gut commensal bacteria has been associated with increased risk of obesity.
· It is unclear to me whether this review discusses the effects of anti-inflammatory diet in modulating gut microbiota and controlling obesity in both children and adults. The references used may or may not be relevant.
· Line 36-37: meaning unclear- "reduced the generation of obesity-suppressing short-chain fatty acids (SCFAs)". This statement should be clarified.
· Line 54-56: Please clearly define “anti-inflammatory” diets here. Several studies have demonstrated that an anti-inflammatory diet (also known as the hypoallergenic/ elimination diet) used as a potential intervention for the treatment of many diseases including allergies, irritable bowel syndrome, inflammatory bowel disease...etc.
· Line 57-60: The novelty of this review is very weak. There are many reviews of its kinds. How this review extends reader understanding of the topic. What this review adds in light of previous reviews? This ought to be explained at the end of the section.
· I understand this is a literature/narrative review, but a method section should be included. It should describe the search terms and types of study designs (e.g., in vivo, in vitro, intervention, RCT, observational studies …etc.) you used, the databases (e.g., PubMed) you searched, the inclusion/exclusion you addressed....etc.
· Line 76-77: “It should be noted that SCFAs….in colonic inflammation. How? This should be expanded to include more details. I would suggest authors referring to this recent review “The Role of Short-Chain Fatty Acids in Mediating Very Low-Calorie Ketogenic Diet-Infant Gut Microbiota Relationships and Its Therapeutic Potential in Obesity”.
· Line 77-78: “several studies….”. This statement should be supported with references.
· Line 85: Please clarify the role of adipokines (e.g., leptin, adiponectin, resistin and ghrelin) here.
· Line 86-88: Meaning unclear- Please revise the statement to make it clear.
· Figure 2: Pro/anti-inflammatory adipokines should be revised to include more details (e.g., IL-8, IL-17, LPS-induced tight junction, zonula occludens-1 (ZO-1), nuclear factor kappa B (NF-kB), Foxp3 Treg, PPAR-y…etc).
· Line 93-108 & Line 110-121: These statements contain little synthesis of the information. For example, how macrophages, neutrophils, B cells, T cells, and others, release the majority of inflammatory molecules in the adipose tissue of obese humans?
· Section 3 and 4 have similar heading.
· Line 368-377: The conclusion is very short- the implications of this review and the directions for future studies should be clearly mentioned.
· English language should be improved in this paper. Several sentences/statements are unclear and should be revised to make it clear.
Round 2
Reviewer 2 Report
Dear Authors,
The paper has significantly improved by these revisions. However, some points are are not sufficiently addressed.
1. Line 58-64: The novelty of this review could be improved. What is new? Authors should clarify why this review is important/significant in light of previous reviews? For example, previous reviews have addressed the......... However, several key knowledge gaps remain to be investigated. This review.....
2. Line 72: Please delete in vivo. The effects of an anti-inflammatory diet in modulating gut microbiota and controlling obesity are highlighted in humans only.
3. Line 73: The duration (2018-2022) is incorrect here. Please check the reference list.